# Combined Robotic Surgery for Double Renal Masses and Prostate Cancer: Myth or Reality?

**DOI:** 10.3390/medicina56060318

**Published:** 2020-06-26

**Authors:** Giovanni Cochetti, Diego Cocca, Stefania Maddonni, Alessio Paladini, Elena Sarti, Davide Stivalini, Ettore Mearini

**Affiliations:** Department of Surgical and Biomedical Sciences, Urology Clinic of Perugia, Perugia University, 06100 Perugia, Italy; giovannicochetti@libero.it (G.C.); stefania.maddonni@live.it (S.M.); alessiopaladini89@gmail.com (A.P.); ele.sarti@libero.it (E.S.); davide.stivalini@gmail.com (D.S.); ettore.mearini@unipg.it (E.M.)

**Keywords:** robotic, prostatectomy, radical prostatectomy, partial nephrectomy, combined robotic surgery

## Abstract

With the widespread use of imaging modalities performed for the staging of prostate cancer, the incidental detection of synchronous tumors is increasing in frequency. Robotic surgery represents a technical evolution in the treatment of solid tumors of the urinary tract, and it can be a valid option in the case of multi-organ involvement. We reported a case of synchronous prostate cancer and bifocal renal carcinoma in a 66-year-old male. We performed the first case of a combined upper- and lower-tract robotic surgery for a double-left-partial nephrectomy associated with radical prostatectomy by the transperitoneal approach. A comprehensive literature review in this field has also been carried out. Total operative time was 265 min. Renal hypotension time was 25 min. Blood loss was 250 mL. The patient had an uneventful postoperative course. No recurrence occurred after 12 months. In the literature, 10 cases of robotic, radical, or partial nephrectomy and simultaneous radical prostatectomy have been described. Robotic surgery provides less invasiveness than open surgery with comparable oncological efficacy, overcoming the limitations of the traditional laparoscopy. During robotic combined surgery for synchronous tumors, the planning of the trocars’ positioning is crucial to obtain good surgical results, reducing the abdominal trauma, the convalescence, and the length of hospitalization with a consequent cost reduction. Rare complications can be related to prolonged pneumoperitoneum. Simultaneous robotic prostatectomy and partial nephrectomy appears to be a safe and feasible surgical option in patients with synchronous prostate cancer and renal cell carcinoma.

## 1. Introduction

In 2018, 1,276,106 new cases of Prostate Cancer (PCa) and 403,262 new cases of kidney cancer were registered worldwide, representing 7.1% of all cancers in men and 2.2% in both sexes, respectively [1]. In patients affected by PCa, the risk of having a synchronous tumor of the kidney and pelvis increases by 6.12 times compared to the general population [2]. The combination of kidney cancer and PCa is not common and is reported to be 0.83%. However, the rate of detection of kidney cancer in patients who had been subjected to staging workup for PCa was significantly increased compared with that expected in the general population, with a Standardized Incidence Ratio (SIR) of 18.19 [3]. Moreover, renal multifocal disease was noted in 15.2% of patients with renal cell carcinoma [4]. Generally, surgical treatment of synchronous kidney cancer and PCa is performed in two stages, and the robotic approach has been established as safe and effective therapy for both tumors. Indeed, compared with open surgery, the employment of the robotic approach provides significant advantages in both kidney cancer and PCa treatment due to the magnification of the operative field and the improvement of surgical accuracy. Concerning PCa, these benefits helped to obtain better functional outcomes in terms of urinary continence and sexual potency [5,6]. The robotic approach used in nephron sparing surgery helps to obtain a lower rate of conversion to open surgery and to radical surgery, shorter warm-ischaemia time, smaller change in estimated GFR after surgery, and shorter length of hospital stay [7]. The use of a minimally invasive surgery can make the solution effective, safe, and with less invasiveness in the case of multi-organ treatment: it could be considered as an option to reduce the length of hospital stays, patient anxiety, and surgical burden [8,9]. Although the combined surgery has not been established as standard treatment for simultaneous PCa and kidney cancer, due to robotic surgery growing, the combined robot-assisted surgery should also be considered. The aim of this study was to report the case of robotic combined surgery as treatment for synchronous PCa and multifocal renal cancer. The secondary aim was to review the literature concerning simultaneous robotic prostatectomy and double-partial nephrectomy. The Ethical Approval was obtained by the CEAS Ethics Committee of Umbria Region (CEAS, no.3193/18). The study was approved on 26 March 2018.

## 2. Case Report

In October 2018, a 66-year-old man with hypertension and no significant prior surgical history showed a serum PSA level of 4.5 ng/mL. At clinical stage T1c, Gleason score 6 (3 + 3) PCa was diagnosed in 11 out of 22 core biopsies, all in the right side of the gland. A total body bone scan was negative for metastasis. An abdominal CT scan showed no evidence of pathologic pelvic lymphadenectomy but highlighted two incidental heterogeneous enhancing solid masses in the left kidney that were suspicious for renal cell carcinoma: one sized 32 mm in the lower pole and the second of 10 mm in the mesorenal site. A 6-cm asymptomatic simple cyst was found in contralateral kidney. Serum creatinine was 1.43 mg/dL. CT imaging of renal masses is shown in Figure 1 and Figure 2.

We performed a robotic, simultaneous double-partial nephrectomy and radical prostatectomy by a transperitoneal approach, using the Da Vinci-Xi Robotic Surgical System (Intuitive Surgical, Sunnyvale, CA, USA).

A robotic, transperitoneal double-partial nephrectomy was performed as the first procedure. Patient positioning was in the right lateral decubitus with a tilt of 30°. An 8-mm trocar was placed above the umbilicus along the left pararectal line for the robotic camera. Pneumoperitoneum of 12 mmHg was established for the other trocars’ placement. An 8-mm trocar for the right arm was placed at the left pararectal line under the subcostal margin and another 8-mm one for the left arm over the left anterior superior iliac spine. The 12-mm AirSeal assistant trocar was placed between the camera port and left robotic arm. Figure 3 shows the trocars’ placement.

The left colon was mobilized by incision along the Toldt line. After identification of the gonadal vein and the ureter, the renal hilum was isolated, and a vessel loop was passed twice around the artery and pulled out extracorporeally parallel to the assistant trocar. Both renal masses were identified on the anterior side of the lower pole of the kidney. Double enucleation with local hypotension was performed according to our technique, previously described: before resection, the vessel loop was tightened in order to apply a progressive occlusion of the arterial lumen easily from the outside and, in this way, obtain a renal hypotension (Figure 4) [10]. The collecting system was not violated. Renorraphy was carried out by sliding clips suture and, then, Floseal Hemostatic Matrix was applied. A pararenal drain was placed. The specimen was located in Endobag.

After robot undocking, the assistant trocar and the right robotic trocar were removed, and port sites were closed. The patient was positioned in a supine, lithotomy, Trendelenburg position. Trocars were placed for the robotic prostatectomy, as shown in Figure 3. The 8-mm camera trocar was now used for the right robotic arm and the trocar in the left iliac fossa was used for the fourth arm. An 8-mm trocar was positioned at the navel for the camera. Another 8-mm robotic trocar was positioned in the right pararectal side. The 12-mm AirSeal assistent trocar was placed laterally between the right anterior superior iliac spine and the left robotic trocar. The robot was then re-docked. Therefore, a full nerve-sparing, robot-assisted, radical prostatectomy was performed according to the Perusia technique (Figure 5) [11]. A drain was posed on the Retzius space. The prostate specimen was put into a second Endobag. Extraction of the specimens was through Mac Burney incision.

Total operative time was 265 min. Total console time was 205 min: 80 min for partial nephrectomy and 125 min for radical prostatectomy. Renal hypotension time was 25 min. Estimated blood loss was 250 mL (partial nephrectomy: 100 mL, radical prostatectomy: 150 mL); no blood transfusion was necessary. No perioperative complications occurred. The catheter was removed on the 6th postoperative day, and the patient was discharged on the 7th. Final pathologic exams demonstrated kidney oncocytoma, papillary adenoma, and pT2b Gleason 6 (3 + 3) prostate adenocarcinoma. All the surgical negative margins were negative. At the 12-month follow-up, no renal or prostatic recurrence occurred, PSA was undetectable, and kidney function was not impaired. Hypertension persisted after surgery.

## 3. Discussion

We reported a case of combined robotic surgery including double-partial nephrectomy and radical prostatectomy as treatment for bifocal renal masses and PCa. Given the incidence of synchronous tumors of the urinary tract, an imaging study of the complete abdomen during the staging of PCa may be advisable to avoid underdiagnosing any synchronous neoplasia, especially renal cell carcinoma.

In our case, considering the bifocality of the renal masses and the clinical features, we decided not to perform a preoperative histological diagnosis using percutaneous renal biopsy and to proceed directly with double-partial nephrectomy simultaneously with radical prostatectomy.

Robotic surgery provides less invasiveness than open surgery with comparable oncological efficacy. The robotic technique helps to maintain the advantages of laparoscopy and to obtain more benefits due to magnification of the operative field through three-dimensional vision and high definition, as well as more accurate movements by EndoWrist^®^ instruments (Intuitive Surgical Inc) with seven degrees of motion. Thanks to these technological evolutions, the same surgical steps of traditional surgery can be reproduced combined with the benefits of a minimally invasive technique, overcoming the limitations of the laparoscopy: an unstable video camera, limited range of instruments’ movements, two-dimensional imaging, and poor ergonomics for the surgeon [12]. Currently, robotic prostatectomy is the standard treatment for PCa. Otherwise, the role of the robotic approach in partial nephrectomy is still debated, even if it allows an easier suture of renal parenchyma and, if necessary, an easier reconstruction of the collecting system. It is increasingly evident that robotic-assisted, partial nephrectomy for multiple ipsilateral renal masses is safe and feasible with minimal damage to renal function [13,14]. Therefore, in cases of synchronous prostate cancer and multifocal renal cell carcinoma, a combined robotic approach could be a valid surgical option. The robotic system facilitates the identification of anatomical structures and better assists the procedure of some complex surgical steps in a narrow space, such as the pelvis. These advantages could facilitate complex combined surgery for synchronous tumors.

We preferred a simultaneous surgery for three reasons: firstly, to avoid a second delayed oncological surgery; secondly, to reduce the risks associated with a double and close anesthesia; thirdly, to reduce stress and anxiety due to a double intervention. Furthermore, this allowed us to treat the patient’s diseases with only one hospitalization and one convalescence.

In 2009, Patel et al. [15] reported the first combined robotic partial nephrectomy and robotic radical prostatectomy for synchronous PCa and renal cell carcinoma, highlighting its feasibility.

In the literature, only 10 cases worldwide of combined robotic surgery for simultaneous RCC and PCa were reported (Table 1, Table 2 and Table 3). To our knowledge, our report is the first case of a combined upper- and lower-tract robotic surgery for a double-left-partial nephrectomy associated with radical prostatectomy.

Combined robotic procedures reduce abdominal trauma, the convalescence, and, therefore, the length of hospitalization with a consequent cost reduction [16,17]. The decrease in the number of applied ports compared to each single procedure minimizes the risk of trocar-positioning injury and also provides a better cosmetic result. The planning of the trocar positioning is crucial to make them usable in both interventions, as well as the extraction site. This advantage is increased by using the Xi system, whose trocars all have a diameter of 8 mm, including the camera port. In our case, the port used for the camera during partial nephrectomy was reused for the bipolar forceps during prostatectomy. In the same way, the port used for the bipolar scissors during the first surgical time was reused for the Prograsp forceps in the second surgical time. In this way, we reduced the risk of potential abdominal traumatism.

However, combined robotic surgery presents disadvantages. The operating time is certainly greater than the operating time of each single procedure. The patient’s general health conditions should help to sustain general anesthesia and pneumoperitoneum for a greater amount of time. The increased operative time may lead to a higher risk of perioperative complications, such as deep venous thrombosis, pulmonary embolism, and even compartment syndrome of the leg [17]. Jung et al. also reported the feasibility of combining bilateral partial nephrectomy and prostatectomy [18]. They showed prolonged anesthesia and operative time and pneumoperitoneum. However, duration of pneumoperitoneum seems to be a significant risk factor in pulmonary mechanics, but it has only a slight effect on overall hemodynamic parameters [19]. In the literature, mean operative time for simultaneous partial nephrectomy and radical prostatectomy was 342 min (240–557). In our case, both procedures were performed in 265 min and no complications related to prolonged pneumoperitoneum occurred. However, the injuries due to prolonged operative time and pneumoperitoneum become significant when the operative time exceeds about 5 h [20].

Moreover, only two postoperative complications were reported in the literature. Boncher et al. described a renal postoperative bleeding leading to selective renal embolization [16]; a ureteral injury occurred during wide excision to ensure a negative surgical margin in patients with locally advanced PCa [21].

Concerning costs, Lavery et al. showed that financial benefits of combining the procedures are substantial with cost savings of over $10,000 in operating-room and postanesthesia-care-unit charges compared with two single procedures. Moreover, when including the additional costs of an additional hospitalization and the associated services (laboratory, pharmacy, and other services), the savings would be even greater [16]. Finally, we estimated a saving of about 5300€ on the surgical procedure by performing combined robotic surgery.

## 4. Conclusions

Our case showed that simultaneous robotic prostatectomy and double-partial nephrectomy appears to be a safe and feasible surgical option in patients with synchronous PCa and renal cell carcinoma. Our findings are in line with what is reported in the literature. However, future large clinical trials are needed to demonstrate the benefits of this combined surgery.

## Figures and Tables

**Figure 1 medicina-56-00318-f001:**
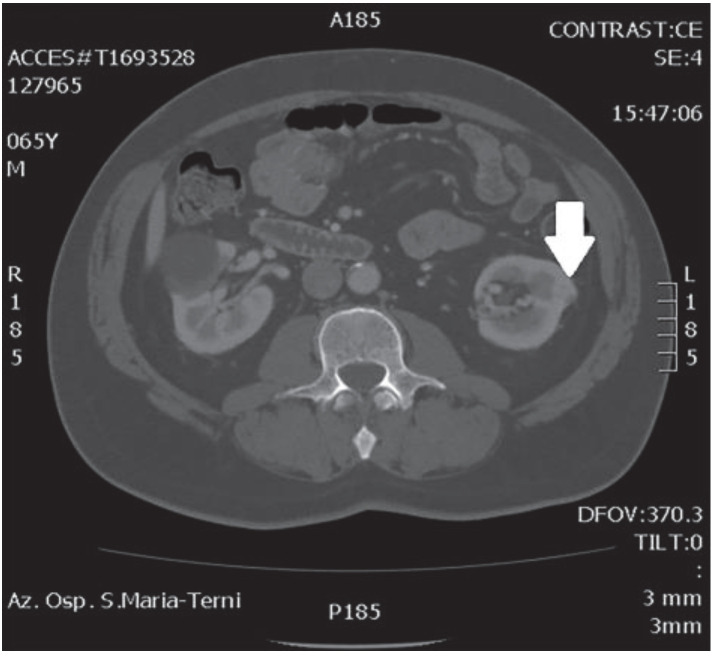
Mesorenal mass.

**Figure 2 medicina-56-00318-f002:**
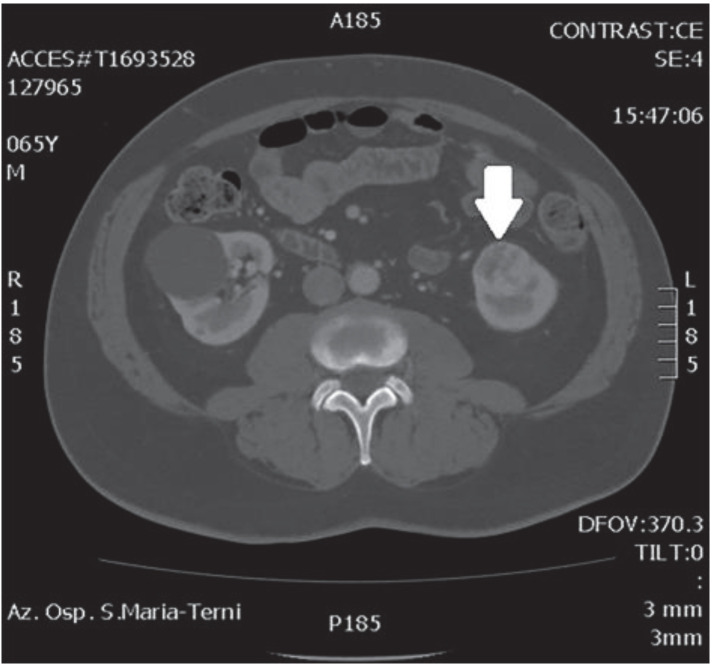
Renal mass of the lower pole.

**Figure 3 medicina-56-00318-f003:**
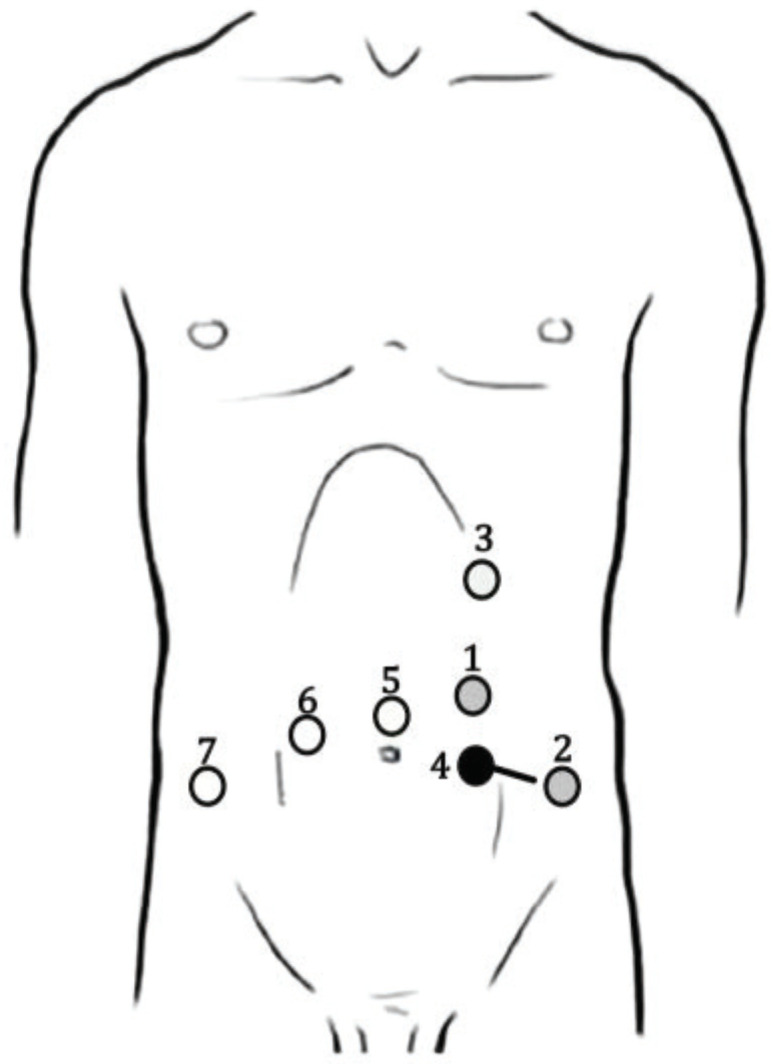
Trocars’ placement during robotic left double-partial nephrectomy and prostatectomy. Black points represent nephrectomy trocars, white points prostatectomy ports, and gray points are trocars used in both operating times. The black line represents the extraction site. 1: Camera port/Bipolar forceps (8 mm); 2: Bipolar scissors/Prograsp (8 mm); 3: Bipolar forceps (8 mm); 4: Assistant port (AirSeal 12 mm); 5: Camera port (8 mm); 6: Bipolar scissors (8 mm); 7: Assistant port (AirSeal 12 mm).

**Figure 4 medicina-56-00318-f004:**
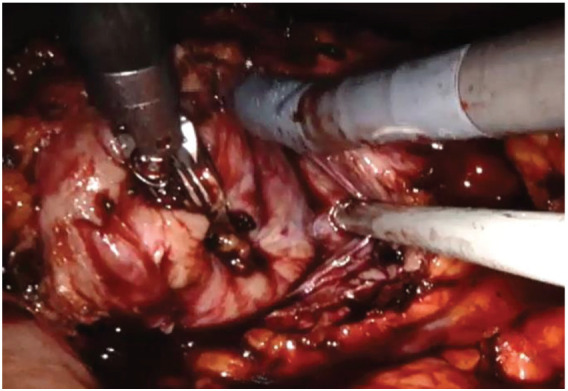
Robotic renal enucleation.

**Figure 5 medicina-56-00318-f005:**
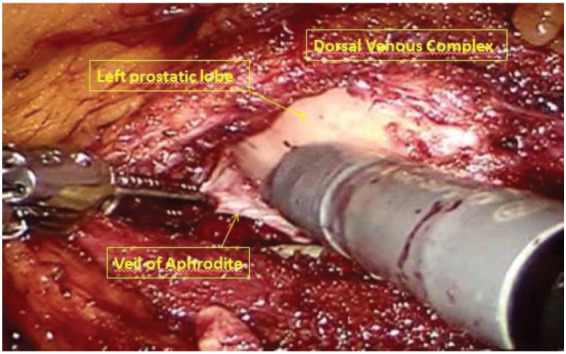
Full nerve-sparing, robot-assisted, radical prostatectomy.

**Table 1 medicina-56-00318-t001:** Preoperative patients’ characteristics.

N.	Authors	Year	Patient				Prostate			Kidney		
			Age	BMI	Comorbidity	ASA	PSA (ng/mL)	Gleason Score	Clinical Stage	Creatinine (mg/dL)	Tumor Size (mm)	Side
1	Patel M.N.	2009	59	-	-	-	21.1	6 (3 + 3)	T1c	1.1	17	Right
2	Boncher N.	2010	49	35	HTN	3	7	6 (3 + 3)	T1c	1.2	50	Right
3	Boncher N.	2010	72	26	HTN, DM	3	29.1	7 (3 + 4)	T2a	1.4	28	Right
4	Lavery H.J.	2010	60				8.4	7 (4 + 3)	T2a	1.0	40	Left
5	Guttilla A.	2011	56	-	No	-	4.8	7 (4 + 3)	T1c	-	20	Left
6	Jung J.H.	2012	62	24.17	-	-	47	8 (4 + 4)	T3b	-	25	Right
											55	Left
7	Jung J.H.	2014	72	28	-	-	0.30	8 (5 + 3)	Salvage	1.1	53	-
8	Jung J.H.	2014	55	24	HTN, DM	-	7.42	8 (4 + 4)	T3a	1.0	27	-
9	Jung J.H.	2014	61	23	HTN	-	61.21	7 (3 + 4)	T3b	1.0	16	-
10	Raheem A.A.	2016	61	25.4	No	-	6.7	6 (3 + 3)	T1c	1.0	16	Right
11	Our patient	2018	66	25	HTN	2	4.5	6 (3 + 3)	T1c	1.43	32	Left
											10	

BMI = body mass index; ASA = American Society of Anesthesiologists; PSA = prostate-specific antigen; HTN = hypertension; DM = diabetes mellitus.

**Table 2 medicina-56-00318-t002:** Intraoperative parameters.

N.	Initial Procedure	OP Time (mins)		Console Time (mins)		Type of Nephr	WIT (mins)	EBL (mL) (Nephrectomy/Prostatectomy)	Complic.	Treatment
		Nephr	Prostat	Nephr	Prostat					
1	Nephrectomy	427		177	158	Partial	24	200 (25/175)	No	-
2	Nephrectomy	120	140	-		Radical	-	150 (50/100)	No	-
3	Nephrectomy	150	138	-		Partial	34	250 (150/100)	PO Bleeding	Angio-embolization
4	Prostatectomy	300		158	120	Radical	-	200 (100/100)	No	-
5	Nephrectomy	-		-		Partial	-	-	-	-
6	Prostatectomy	557		116 (right)	164	Partial	24 (right)	500 (300/200)	No	-
				88 (left)			27 (left)			
7	Prostatectomy	136	206	78	110	Partial	33	150 (50/100)	No	-
8	Prostatectomy	123	144	89	111	Partial	24	800 (150/650)	No	-
9	Nephrectomy	150	330	103	272	Partial	35	700 (50/650)	Ureteral injury	Ureteroneocystostomy
10	Prostatectomy	240		71	61	Partial	19	300 (250/50)	No	-
11	Nephrectomy	110	155	80	125	Double partial	25	250 (100/150)	No	-

OP = operation; WIT = warm ischemic time; EBL = estimated blood loss.

**Table 3 medicina-56-00318-t003:** Postoperative Results.

N.	Creatinine (mg/dL)	Length of Stay (days)	Prostate			Kidney		Length FU (months)	Evidence Recurrence
			Gleason Score	Pathologic Stage	PSM	Pathology/Stage	PSM		
1	-	2	7 (4 + 3)	-	No	Clear cell, grade 2/T1a	No	4	No
2	1,8	3	7 (3 + 4)	T2aN0M0	No	RCC, grade 3/-	No	10	No
3	1,0	2	7 (3 + 4)	T3bN0M0	No	RCC, grade 2/-	No	6	No
4	1,3	2	7 (3 + 4)	T2cN0M0	No	Clear cell, grade 2/T1a	No	-	-
5	-	-	-	-	-	-	-	-	-
6	1,31	-	9 (4 + 5)	-	Yes	Clear cell, grade 2/T1a	-	2	Prostate
						Clear cell, grade 3/T3a	-		
7	1,3	7	0	T0N0M0	No	Clear cell, grade 2/T1b	-	18	-
8	1,1	7	8 (3 + 5)	T2bN0M0	No	Clear cell, grade 3/T1a	-		
9	1,4	13	8 (3 + 5)	T4aN0M0	Yes	Clear cell, grade 2/T1a	-		
10	-	4	7 (3 + 4)	-	No	Clear cell, grade 2/T1a	No	-	-
11	1,4	9	6 (3 + 3)	T2b	No	Oncocytoma Papillary adenoma	-	1	No

PSM = positive surgical margin; FU = follow-up.

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
