# Peer review of "Combined Robotic Surgery for Double Renal Masses and Prostate Cancer: Myth or Reality?"

_1010-660X, 2020, doi:10.3390/medicina56060318_

Round 1

Reviewer 1 Report

Review of Manuscript ID medicina-783548
Title: Combined robotic surgery for double renal masses and prostate cancer: myth or reality?

Review comments and suggestions for authors:

  • In the CT images it seems like the right kidney has large cyst. It has to be identified as comorbidity. Please clarify if the hypertension is related to the renal mases or not.
  • One or two intraoperative images can be relevant to the presentation of the surgical technique, especially for the partial nephrectomy.

Reviewer 2 Report

This is a great honor to review in Medicina.

After thorough reading of the present case report, I raised several major questions to authors.

  1. First of all, what is the clinical implications of the present case report? Two surgery in one schedule? Why is it important? In the present case report, author described that after robot-assisted partial nephrectomy completed, patient was re-positioned, re-drapped, and even trocar-cannulated for robot-assisted prostatectomy. Is two combined surgery in one schedule is better two separate scheduled surgery? I think it is not sure yet and author should describe that why they do such a combined surgery first, not how they do such a combined surgery. 
  2.  The advantage of combined surgery is not confirmed. Although authors described that combined procedures reduce the abdominal trauma, therefor the time of hospitalization with a consequent reduction in costs, it is still controversial and such a conculsion cannot be drawn from the present case report. 
  3.  What is the incidence of combination of RCC and PC? How many cases (even in worldwide..) of combined surgery of partial nephrectomy and prostatectomy? 

Round 2

Reviewer 2 Report

The author did respond to all reviewer's comments appropriately. Although the technique would not be technically novel or unique, it could be considered as an option to reduce hospital stay, patient anxiety, or surgical burden. As is robotic surgery growing, the combined robot-assisted surgery also should be considered.

Author Response

Response to Reviewer 2 Comments

Point 1: The author did respond to all reviewer's comments appropriately. Although the technique would not be technically novel or unique, it could be considered as an option to reduce hospital stay, patient anxiety, or surgical burden. As is robotic surgery growing, the combined robot-assisted surgery also should be considered.

Response 1: We thank the reviewers for their advice and for allowing us to improve our manuscript. We completed the introduction by specifying the background of our case report by adding specific references. We have carried out a revision of the English language by a native English speaking colleague. We agree with you that the technique described in this case report is a valid alternative to traditional surgical techniques. The observed advantages are in line with those reported in the literature. However, future large clinical trials are needed to demonstrate the benefits of this combined surgery.